# A New Linear Relation for Estimating Surface Broadband Emissivity in Arid Regions Based on FTIR and MODIS Products

**Huoqing Li** [1,2]**, Zonghui Liu** [1]**, Ali Mamtimin** [1,]*** , Junjian Liu** [1]**, Yongqiang Liu** [3]**, Chenxiang Ju** [1]**, Hailiang Zhang** [1] **and Zhibo Gao** [4]

1   Institute of Desert Meteorology, China Meteorological Administration, Urumqi 830002, China;
    lihq@idm.cn (H.L.); liuzh@idm.cn (Z.L.); liujj@idm.cn (J.L.); jucx@idm.cn (C.J.); zhanghl@idm.cn (H.Z.)
2   College of Atmospheric Sciences, Lanzhou University, Lanzhou 730000, China
3   College of Resource and Environment Science, Xinjiang University, Urumqi 830002, China; liuyq@xju.edu.cn
4   College of Global Chang and Earth System Science, Beijing Normal University, Beijing 100088, China;
    gaozhibo@mail.bnu.edu.cn
*   Correspondence: ali@idm.cn; Tel.: +86-0991-2652429

**Abstract:** Broadband emissivity is a crucial parameter for calculating the radiation budget, still, it adopts a constant value in land surface models due to a lack of adequate observations. Arid regions have complex underlying surfaces and estimations of the broadband emissivity in such areas suffer from high spatial variation and uncertainty. Here, we propose a novel method for estimating broadband emissivity in the 8–14 μm range based on Fourier-transform infrared spectroscopy (FTIR) observations, moderate resolution imaging spectrometer (MODIS) emissivity, the leaf area index (LAI) and reflectance products. The proposed method exploits FTIR observations, MODIS single-channel emissivity, reflectance and the LAI to fit a linear regression of the broadband emissivity, so the optimal equation includes emissivity, reflectance and the LAI, with an $R^2$ and root-mean-squared error of 0.942 and 0.08. Then we used the proposed method to generate a broadband emissivity map of Northwest of China, the broadband emissivity estimated by the method showed higher variations and finer distribution in arid areas and sparsely vegetated regions compared to data from the global land surface satellite and land model. An analysis of the relationship between the broadband emissivity, land-use type and soil moisture found an existing but not linear relationship, which indicated that the relationship was complicated under the inhomogeneous surface of wetness and vegetation. In conclusion, our results suggest that the proposed method can accurately estimate the broadband emissivity in arid regions. In future research, we will test the data in a land model.

**Keywords:** broadband emissivity; FTIR; MODIS; LAI; arid region

## 1. Introduction

The land surface thermal-infrared (TIR) broadband emissivity (BBE) is a key parameter used to calculate the Earth's surface energy budget [1–5]. The emissivity is defined as the ratio of energy emitted from natural material to that from an ideal blackbody at the same temperature [6]. Typically, the broadband wavelength of BBE ranges from 8 to 14 μm, which is used to determine the long-wave radiation in the atmosphere from the surface and has been used in the land surface models and general circulation models [7–9]. However, due to the lack of reliable regional emissivity observations, the BBE used in the land surface model was derived from parameter tables by a look-up table and defined according to each land-use type. Thus, every kind of land use matches a constant, or simple schemes are adopted in surface modeling frameworks [10–12]. For example, the unified Noah land model of the National Center for Atmospheric Research Applications Laboratory uses a look-up table from VEGPARM.TBL to match the vegetation categories of each cell to acquire the land use type. It then calculates the emissivity according to the areal fractional coverage of

green vegetation [13–15]. This approach will cause an obvious difference between estimated and real BBE. Previous sensitivity studies by a climate model indicated that a decrease of BBE by 0.1 would increase the simulated surface and air temperatures by about 1 K and decrease the upward radiation more by than 8 W·m$^{-2}$ over the Sahara Desert. One feasible way to overcome this lack is to use remote sensing data to retrieve the BBE. Past research suggested that satellite-derived BBE could improve the performance of climate models [6]. In addition, remote sensing data with a higher spatial and temporal resolution can be helpful for land surface process studies at the regional scale and serve as medium-scale data to validate coarse-resolution data, thereby improving our understanding of land-atmosphere interactions [16].

In recent decades, great efforts have been made to determine BBE from remote sensing thermal infrared (TIR) data [17]. Three such methods can be employed to estimate regional land surface BBE. The first method is classification-based and involves measuring the BBE from each type of land surface in a laboratory, then assign the measured value for the type of land surface. Wilber et al. mapped the global BBE with 10′ × 10′ spatial resolution, the global surface was divided into 18 types and each cell was filled with a constant BBE that was measured by spectral data [18]. With the development of remote sensing technology, fractional vegetation coverage (FVC) and normalized difference vegetation index (NDVI) have been used to estimate BBE. This approach has been widely used to retrieve land surface temperatures by predetermining surface emissivity [19]. This method obtains a highly accurate static BBE value of specific surface types, but one disadvantage is that it cannot represent the inhomogeneous surface and it is hard to obtain high-quality BBE. The second method involves converting the emissivity from TIR narrowband to broadband by adopting a linear combination. Previous studies proposed an approach to using the moderate resolution imaging spectroradiometer (MODIS) data or advanced space-borne thermal emission and reflection radiometer (ASTER) narrowband emissivity data to estimate the BBE [7], Ogawa et al. mapped the North African BBE (8–13.5 μm) using the ASTER emissivity product (90 m), the range of the BBE was found to be between 0.85 and 0.96 for the desert [3]. Jin et al. converted MODIS TIR narrowband emissivities into BBE, deriving the estimated BBE improved the performance of global climate models over desert areas [6]. As the number of satellites with TIR channels increases, this method has been extended to Landsat, Fengyun Visible Infrared Imaging Radiometer Suite (VIIRS) and Spinning Enhanced Visible Infra-Red Imager (SEVIRI) satellite Data [19–22]. Based on MODIS narrowband emissivity products, the second method has been used to produce a global land surface satellite (GLASS) BBE dataset for the years from 1981 to 2010 [23–25].

The third method directly establishes a relationship between field-measured spectral data and the narrowband emissivity to estimate the BBE [26,27]. This method has practical physics-based regression with the narrowband emissivity of the remote sensing data and it is correlated with the real surface emissivity. Li et al. explored using this method based on portable Fourier Transform InfraRed thermal spectroscopy (FTIR) and MODIS products to estimate emissivity in the Taklamakan Desert (TD) and it examined whether this approach could achieve higher accurate BBE than others, the range of the estimated BBE value range from 0.89 and 0.91 over the desert [27]. Aynigar et al. figured out that the GLASS data obviously overestimated the BBE across the TD. They performed coefficients optimization of the GLASS broadband emissivity based on FTIR and MODIS data over the TD, which estimated that BBE better agrees with field-measurements [28]. Previous studies suggest that the third method is more suitable for BBE estimation of non-vegetated arid regions, but areas with sparse vegetation should not be ignored, so the method still needs to be improved for the entire arid region.

The accuracy of the first method of BBE retrieval is poor due to each type of land surface has noticeable differences that result in uncertainty. Bare soil, for example, affects BBE estimations owing to the soil texture, color and moisture. The ASTER narrowband emissivity indicated that soil emissivity varies from 0.86 to 0.98 [29]. If we adopt a default soil BBE value, the errors will exceed 0.1, which amplifies errors in radiation temperature in the surface

radiation budget [10]. Thus, the first method is not acceptable for retrieving BBE estimations in a model. The second method relies on different remote sensing products. These data have different spatial or temporal resolutions. For example, the ASTER revisiting period is 16 days, and the MODIS revisiting period is 8-day. Thus, the method is limited [7] and cannot be extended to vegetated areas. Furthermore, MODIS NDVI products have poor data quality over the vegetation-covered area, so the BBE derived from NDVI is incorrect in vegetated regions [30]. Semi-empirical regression methods are often highly efficient at retrieval and easy to implement. Thus, a high-quality vegetation parameter should be adopted. Carlson et al. indicated that the leaf area index (LAI) has a high correlation with the BBE and described a linear regression equation to estimate the BBE from Landsat data. This approach performed well in vegetated areas [31]. Generally, surface spectral data is effective and can be used to acquire the BBE directly. Liu used FTIR surface spectral measurements of the Taklimakan Desert and calculated the BBE [32]. Cheng et al. performed field measurements validated by MODIS land surface emissivity products (MOD11B1) over the hinterland of the Taklimakan Desert [26]. The field-measured spectral emissivity is consistent with MODIS single-channel narrowband emissivity. Unfortunately, the FTIR measurement method merely aims at point observations and cannot obtain regional BBE estimations. Remote sensing data could be used to measure regional narrowband emissivity but such data cannot represent broadband spectral emissivity [33,34]. The three methods mentioned above exclusively use remote sensing and spectral library data and land surface characteristics vary considerably. Li and Liu used the MODIS narrowband emissivity and spectral data measured by FTIR fitted with a linear regression equation to assess the BBE of the Taklimakan Desert. Although their equation represented the BBE of the desert well and with high accuracy, it could not reflect the BBE in vegetated areas and they did not establish the relationship with NDVI or LAI.

Therefore, to mitigate the variation and uncertainty in land surface emissivity estimations in arid areas, we propose a method to fit the linear regression equations to estimate the BBE. The equations include the MODIS narrowband emissivity, LAI products and field-measured spectral data. The proposed method exploits the advantages of both FTIR and remote sensing, and we consider vegetated areas to improve the accuracy of the BBE produced from MODIS data.

The structure of this paper is arranged as follows. The introduction of the study area, remote sensing data, spectral measurements and BBE calculations are described in Section 2. The physical mechanism and proposed method are introduced in Section 3. The results and analysis are presented in Section 4. A discussion is provided in Section 5 and the main conclusions are summarized in Section 6.

## 2. Study Area and Data

### 2.1. Study Area

Arid lands are commonly defined as regions in which the annual potential evapotranspiration (PET) greatly exceeds annual precipitation (P) [35–37]. Driven by the underlying trend of global warming, this region is very sensitive to both climate change and land use/land cover change [38,39]. This research focused on the arid land region in northwest China, where land cover is dominated by bare soil and sparse vegetation, as shown in Figure 1a. Our field-measurements were mainly taken in the Taklamakan Desert (TD), the Gurbantunggut Desert (GD) and the Turfan Basin (TB). These regions have an extremely arid climate. Among them, TD and TB average annual precipitation 10~38 mm, while the average annual evapotranspiration is over 3200 mm, the soil texture of TD and TB are sand and dark brown gravel (Figure 1b,c) and the hinterland of TD and TB without vegetation [40,41]. GD has a mean annual precipitation of 135 mm, but its potential evaporation is more than 2000 mm, where land use is covered by sparse needle-leaved scrub (e.g., Sacsaoul, Rose willow) as shown in Figure 1d, where soil texture is mostly sand and clay [42]. The arid land is a sensitive region to climate change response, emissivity is an important parameter for determining the surface radiation budget in climate, weather and hydrological models [3], the higher accuracy emissivity of arid land contributes to

improving the performance of models. This study proposes a new method based on FTIR spectral measurement and MODIS products to estimate broadband emissivity, and this work has a positive significance for climate prediction and simulation in the arid area.

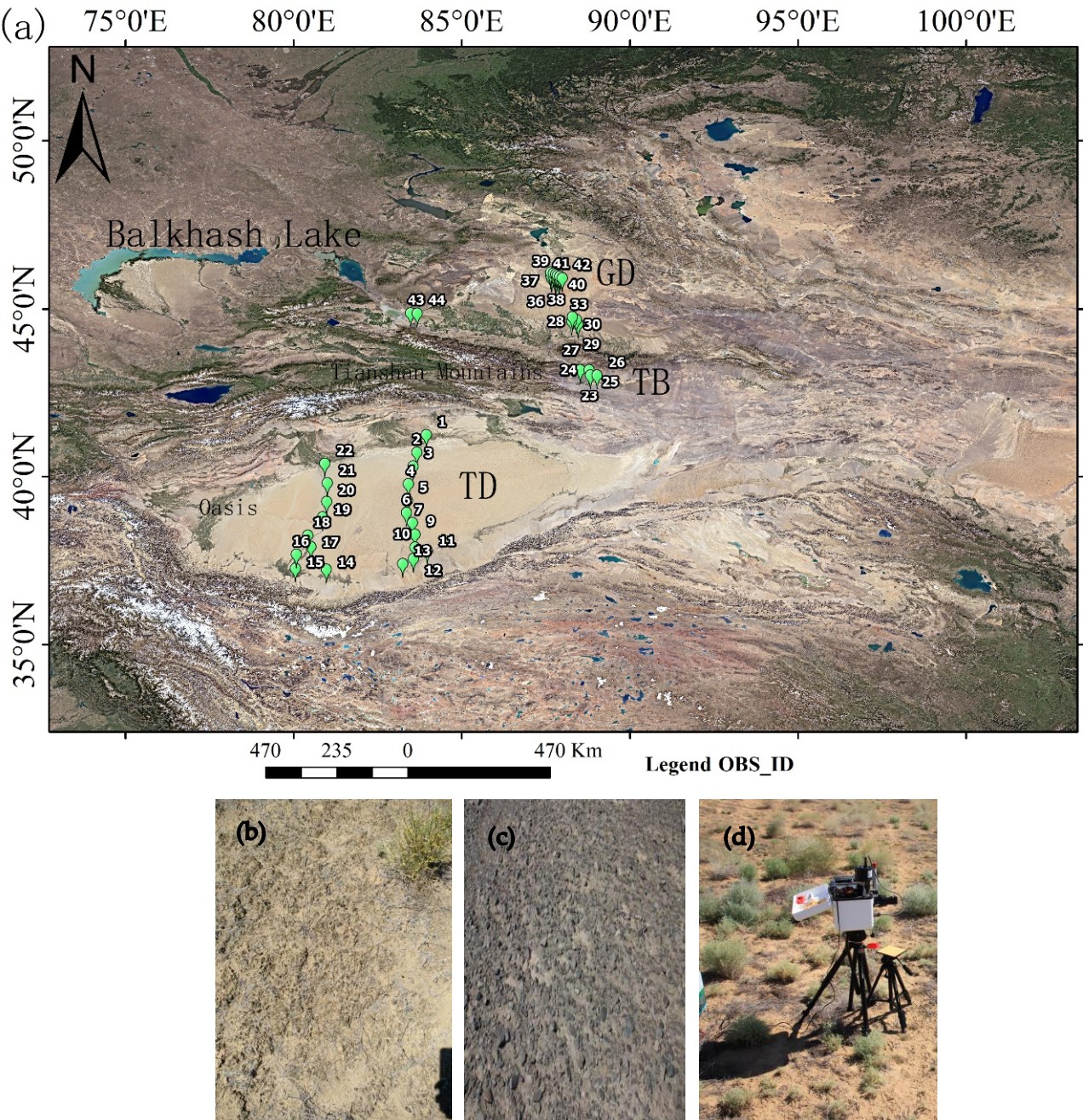

**Figure 1.** Field-measured sites of FTIR arid region from Google Earth (**a**) and surface picture (**b–d**).

### 2.2. Remote Sensing Data

The proposed method needs narrowband emissivity, reflectance and LAI products. The narrowband emissivity is derived from MODIS land surface temperature and emissivity (MOD11B1) products (https://lpdaac.usgs.gov/products/mod11b1v061/) (accessed on 10 November 2020). The data are tile-based, and grids in the sinusoidal projection are produced daily at 5 km. The revisiting period is daily [17]. The MOD11B1 product is produced using a physically-based day/night algorithm. It consists of 14 equations for the solution of 14 land-surface and atmospheric parameters based on the day/night observations of the seven infrared

MODIS bands (bands 20, 22, 23, 29, 31, 32 and 33) [43]. We collected bands 29 (8.40–8.70 μm), 31(10.78–11.28 μm) and 32 (11.77–12.27 μm) because these bands emissivity values have a high relationship with the BBE. MODIS band 31 is expected from laboratory data [44–46], and these bands' wavelengths are closer to the BBE wavelength in the range of 8 to 13.5 μm. In addition, MODIS channel 7 (MOD09A1) (https://lpdaac.usgs.gov/products/mod09a1v006/) (accessed on 11 November 2020) was chosen because it has a high correlation with the broadband emissivity, and it is highest of all among the MODIS reflective channels [2]. The surface reflectivity of soil and rock is determined by their mineral composition, as is their emissivity. Generally, quartz-rich ($SiO_2$) sand has higher reflectivity and a lower emissivity but mafic minerals with lower $SiO_2$ content generally have lower reflectivity and a higher emissivity. Cheng et al. proposed a disaggregation approach that utilizes the established BBE–reflectance relationship to estimate high spatial resolution BBE for bare soils from Landsat surface reflectance data [4]. The LAI is a dimensionless quality that characterizes plant canopies. The LAI is one of the plant biophysical factors that affect the canopy spectral reflectance of plants and directly affects the emission of radiation [47]. The LAI products used here comprise MODIS MOD15A2H products (https://lpdaac.usgs.gov/products/mod15a2hv006/) (accessed on 18 November 2020). These data have the same spatial resolution as MOD11B1.

To evaluate the performance of our method, we chose the global land surface satellite (GLASS) BBE products for comparison and analysis. GLASS products have been widely used in remote sensing (http://www.glass.umd.edu/) (accessed on 10 November 2020) [24]. Nevertheless, we used the global land cover map GLC2015 from the European Space Agency (ESA) climate change initiative project (http://maps.elie.ucl.ac.be/CCI/viewer/download.php) (accessed on 16 November 2020), which has a high accuracy classification beyond 74.58% and we applied it to a numerical weather forecast model [48,49]. The land cover data were used to analyze the feature of BBE distribution and land cover. Soil moisture data were used to interpret the relationship between the BBE estimations. The soil products of Soil Moisture Active Passive (SMAP) Level-4 radiometer data (https://cmr.earthdata.nasa.gov/search/concepts/C1920755724-NSIDC_ECS.html) (accessed on 15 November 2020) were employed here [50] to investigate the characteristic distributions between soil moisture and BBE.

## 3. Spectral Data and BBE Calculations

### 3.1. Spectral Data Measurements

We conducted the field measurements in June 2018 and the entire observation period lasted for three weeks, the observation period was sunny and there was no precipitation in the observation area before and after one week, so the land surface almost without variation. The spectral data derived from the TD, TB and GD, and the date of the field acquisitions of each region are 4 June to 10 June; 13 June to 15 June; 18 June to 22 June 2018, respectively. These regions are the typical arid areas, with widely distributed desert, sparse vegetation. The observation sites are marked in Figure 1a. The observation path crossed through the entire arid region. The Model 102 Portable FTIR Spectrometer (Model 102F), powered by a lead-acid battery, and a Labsphere gold plate were used to measure the spectral data under a clear sky. The thermal emission at a wavelength spectral range of 2 to 16 μm was measured. It was detected at a spectral resolution range of 2–24 cm$^{-1}$, with an indium antimonide (InSb) detector. The emissions at the wavelength ($\lambda$) from 8–14 μm were measured with a mercury cadmium telluride (HgCdTe) at 4 cm$^{-1}$ resolution, with a standard observation deviation of less than 1% [51]. The radiance of the instrument was calibrated by a blackbody, jointed to the FTIR and controlled by the observed temperature. The operating temperature was between air temperature and the land surface temperature. Site measurements were repeated three times and were finished in 10 min to avoid uncertainty and error. Most of the spectral measurements are located in areas with barren soil that were sparely vegetated, because measuring the emissivity of sparsely vegetated to nearly fully vegetated surfaces in the field is technically challenging [52]. The details of the observations and field measurements are described by Liu et al. [34].

The emissivity spectrum was derived from the radiometric measurements by the iterative spectral smooth temperature and emissivity separation algorithm [53]. Figure 2 shows the derived emissivity spectra curve in the spectral range of 8 to 14 μm over three types of arid surfaces and plotted MODIS narrowband emissivity values denoted by circles in bands 29, 31 and 32 in the same position, MODIS emissivity are very close to the measured at the specified wavelength. The emissivity of the driest Taklimakan Desert has the lowest value, especially when the wavelength ranged between 8 and 12 μm. The Turpan Basin is covered by dark brown gravel and its emissivity value is slightly greater than that of the TD. The GD is covered by sparse scrub and has a higher soil moisture, so has, comparatively, the highest emissivity among three regions.

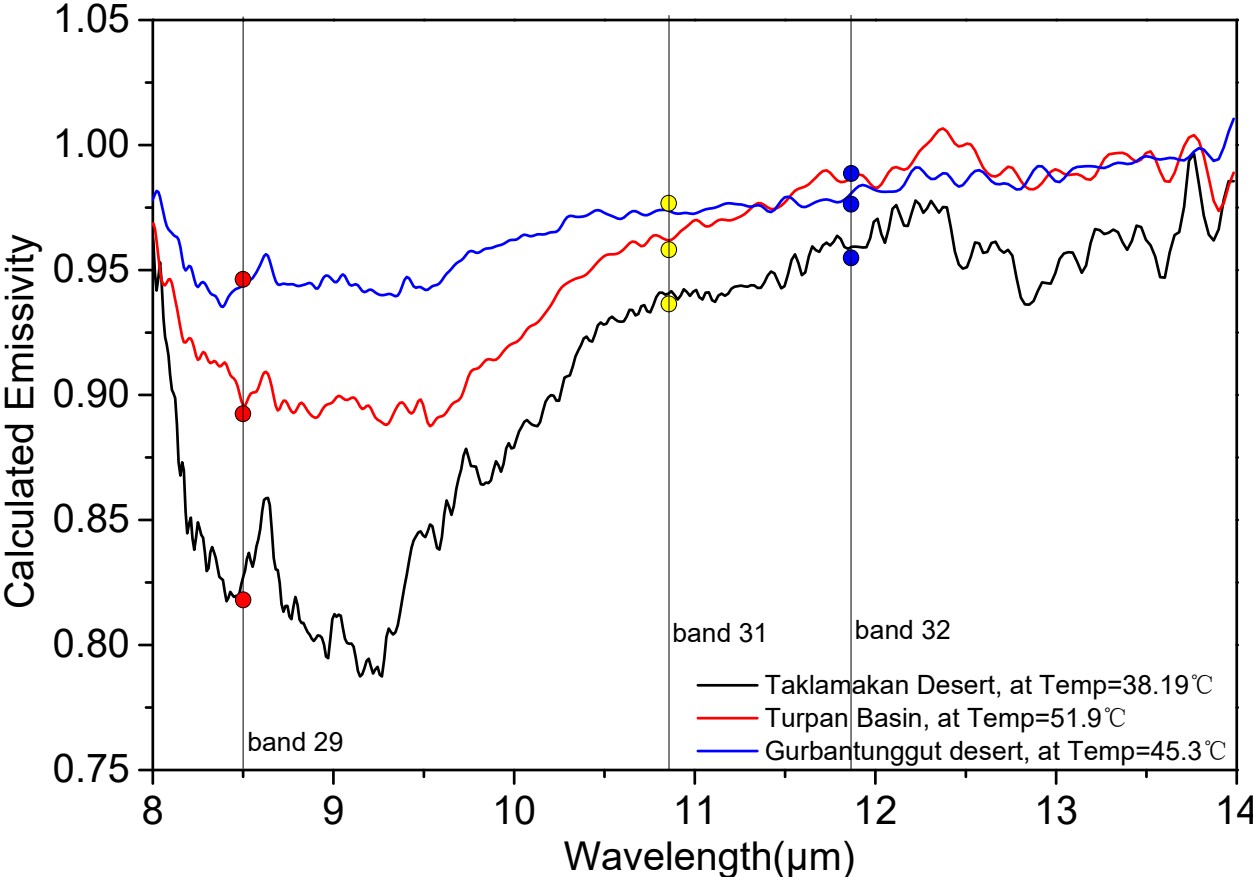

**Figure 2.** Field-measured emissivity spectra of desert, barren soil and gravel. Land surface sites the over Northwest of China arid region. The circles denote the emissivity values from MODIS bands 29 (red), 31 (yellow) and 32 (blue). These three spectral observation points are respectively sites 7, 25 and 41 in Figure 1a.

### 3.2. Converting Spectral Data to BBE

Land surface emissivity values are required but not spectral curve values in the 8–14 μm band [54,55]. The formula for converting wide-band surface emissivity spectra to surface emissivity $\varepsilon$ is as follows:

$$\varepsilon = \frac{\int_{\lambda_1}^{\lambda_2} \varepsilon_s(\lambda) B(\lambda, T_s) d\lambda}{\int_{\lambda_1}^{\lambda_2} B(\lambda, T_s) d\lambda},$$ (1)

where $\lambda_1$ and $\lambda_2$ are the wavelength range of the integral equation—namely, the thermal infrared atmospheric window spectral wavelength range of 8 to 14 μm [55], here, $\varepsilon_s(\lambda)$ and

$B(\lambda, T_s)$ in the equation are continuous functions. To facilitate the calculation, the integral equation is discretized as:

$$\varepsilon = \frac{\sum\limits_{\lambda=\lambda_1}^{\lambda_2} \varepsilon_s(\lambda) B(\lambda, T_s)\delta\lambda}{\sum\limits_{\lambda=\lambda_1}^{\lambda_2} B(\lambda, T_s)\delta\lambda}. \tag{2}$$

To improve the accuracy, the wavelength range of 8 to 14 μm is divided into 375 intervals of $\delta\lambda$ in the calculation. Therefore, we use a MODIS thermal infrared narrowband emissivity, which can be briefly described as

$$\varepsilon_i = \frac{\int\limits_{\lambda_{i1}}^{\lambda_{i2}} f_i(\lambda)\varepsilon_\lambda B_\lambda(T)d\lambda}{\int\limits_{\lambda_{i1}}^{\lambda_{i2}} f_i(\lambda)B_\lambda(T)d\lambda}, \tag{3}$$

where $fi(\lambda)$ is the spectral response function. According to the Equations (1) and (3), the broadband emissivity can be represented by different thermal infrared narrowband emissivity linear combinations [56]:

$$\varepsilon_{\lambda_1-\lambda_2} = \frac{\sum\limits_{i=1}^{n} \int\limits_{\lambda(i)}^{\lambda(i+1)} \varepsilon_\lambda B_\lambda(T)\mathrm{d}\lambda}{\int\limits_{\lambda 1}^{\lambda 2} B_\lambda(T)\mathrm{d}\lambda} = \sum\limits_{i=1}^{n} g_i \varepsilon'_i \approx \sum\limits_{i=1}^{n} g_i \varepsilon_i, \tag{4}$$

where $\varepsilon_i$ and $g_i$ can be expressed respectively as

$$g_i = \frac{\int\limits_{\lambda(i)}^{\lambda(i+1)} B_\lambda(T)\mathrm{d}\lambda}{\int\limits_{\lambda 1}^{\lambda 2} B_\lambda(T)\mathrm{d}\lambda}. \tag{5}$$

Conceptually, $g_i$ is the coefficient of combination, which has a relationship with the thermal radiation luminance of the blackbody and is independent of the emissivity of a single band [57]. Based on these physical theories, we employ the thermal infrared bands of emissivity from MODIS products to estimate the BBE. The reflectance of MODIS channel 7 is sensitive to and fraction of vegetation and soil moisture; additionally, the emissivity has a relationship with the soil texture, organic matter and soil moisture, previous studies have demonstrated an empirical relationship between the BBE and reflectance [53], so the reflectance also reflect emissivity variation [2]. Moreover, the LAI has a high relationship with the BBE over vegetated areas [58]. The emissivity of MODIS bands 30 is absorbed by ozone and this band has considerable uncertainty. Arid regions have large areas covered by barren land and sparse vegetation, with high reflectance and low emissivity. We propose three equations that involve the MODIS narrowband emissivity, band 7 reflectance and LAI products in view of the above situation. These three equations are expressed as follows:

$$\varepsilon_{\mathrm{BB}} = a \cdot \varepsilon_{29} + b \cdot \varepsilon_{31} + c \cdot \varepsilon_{32}, \tag{6}$$

$$\varepsilon_{\mathrm{BB}} = a \cdot \varepsilon_{29} + b \cdot \varepsilon_{31} + c \cdot \varepsilon_{32} + d \cdot \alpha_7, \tag{7}$$

$$\varepsilon_{\mathrm{BB}} = a \cdot \varepsilon_{29} + b \cdot \varepsilon_{31} + c \cdot \varepsilon_{32} + d \cdot \alpha_7 + e \cdot LAI, \tag{8}$$

where $a$, $b$, $c$, $d$ and $e$ are regression coefficients; $\varepsilon_{29}$, $\varepsilon_{31}$ and $\varepsilon_{32}$ are the narrowband emissivity of MODIS bands 29, 31 and 32, respectively; $\alpha_7$ is the MODIS channel 7 reflectance and LAI represents the MODIS LAI product. Equation (7) comprises the narrowband emissivity of MODIS bands 29, 31 and 32. Equation (8) adds the MODIS channel 7 reflectance based on Equation (7). Equation (9) involves MODIS LAI products based on Equation (8). The regression coefficients are the key to producing the regional BBE used to measure sites that match the value from MODIS products that fit the linear regression equation to determine the coefficients.

## 4. Results and Analysis

### 4.1. BBE Estimation Equation

Based on the proposed equations, we employed 44 in situ BBE measurements with the value extracted from MODIS emissivity, reflectance and LAI products to calculate the regression coefficients as follows:

$$\varepsilon_{\mathrm{BB}} = 0.121\varepsilon_{29} + 0.462\varepsilon_{31} + 0.523\varepsilon_{32}, \tag{9}$$

$$\varepsilon_{\mathrm{BB}} = 0.08\varepsilon_{29} + 0.485\varepsilon_{31} + 0.536\varepsilon_{32} - 0.152 \cdot \alpha_7, \tag{10}$$

$$\varepsilon_{\mathrm{BB}} = 0.07\varepsilon_{29} + 0.484\varepsilon_{31} + 0.436\varepsilon_{32} - 0.079 \cdot \alpha_7 + 0.176 \cdot LAI. \tag{11}$$

These three equations are assessed in Table 1. The $R^2$ and RMSE of Equation (9) are 0.83 and 0.17, respectively. The Equation (9) produced BBE exclusively from emissivities of MODIS bands 29, 31 and 32 is not incredibly accurate because its lack of surface reflectance would lead to cannot reflect the soil textures and minerals over complex underlying surfaces. Equation (10) adds the reflectance, and thus the $R^2$ is reduced by 8.4% compared to Equation (9). Reflectance provides more land use information and helps to improve the BBE estimation accuracy. Equation (11) includes both MODIS channel 7 reflectance and LAI products, the reflectance could represent the soil texture variation in spatial, and also LAI can reflect vegetation distribution well. Its determination coefficient $R^2$ was 0.94, and the RMSE was reduced to 0.08 compared to Equation (11). Figure 3 shows a scatterplot of BBE estimated by Equation (11) and in situ field-measured BBE, it illustrates the estimated BBE is close to the observed in the BBE value range from 0.90 to 0.92. The assessment results indicated that the uncertainty of Equation (11) is about ±0.006. Obviously, a significant linear relationship existed between the BBE of estimated and field-measured. From the above statistics, Equation (11) is the best estimation equation, therefore, we recommend Equation (11) as the optimal approach for estimating the BBE in arid regions. In general, the new method is an attempt to overcome the uncertainty in estimating BBE in arid regions [30]. Notice that the LAI was normalized before use, and we limited the estimated BBE in (0.8~1.0) for regional estimations.

**Table 1.** Accuracy of the three proposed regression equations.

| Statistic Index | Equation (9) | Equation (10) | Equation (11) |
|:---:|:---:|:---:|:---:|
| $R^2$ | 0.83 | 0.88 | 0.94 |
| RMSE | 0.17 | 0.1 | 0.08 |
| Bias | 0.14 | 0.09 | −0.007 |

### 4.2. Estimating BBE of Arid Region

We used regression (11) with MODIS data (MOD11B1, MOD09A1, MCD15H2) acquired over the Northwest of China for estimating the BBE. The above MODIS products covered Central Asian arid regions (tile: H23~26, V3~5) and were collected from May to August in 2018. The products were pre-processed by MRT (MODIS Reprojection Tool). We transferred the projection to geographic coordination and reformatted the data from HDF (Hierarchical Data Format) to TIFF (Tagged Image File Format). Then we used ArcGIS to generate the arid-region BBE map based on Equation (11) of June 16, 2018, as shown in

Figure 4. The BBE ranged between 0.89 and 0.99 with more detail at the spatial variation distribution. The majority lower BBE with most range from 0.89 to 0.92 over bare soil surface (denoted in blue), the dominant regions such as the TD and the GD. The higher BBE value is between 0.96 and 0.98, most located over oases where major land-uses are cropland, prairies and forests. The BBE map describes detailed spatial variation over vegetated areas and vegetation–desert transition zones, where major land cover type is open shrub, the ranges BBE from 0.94 to 0.96 over sparsely and partially vegetated (denoted in yellow to orange). The highest BBE is nearly 0.99 over water bodies such as lakes or reservoirs such as Balkhash Lake (denoted red). The BBE values range from 0.95 to 0.99 in Kazakhstan, where the land cover is sparse grassland, rain-fed croplands and sparse vegetation. From the above results BBE was closely related to the land cover types.

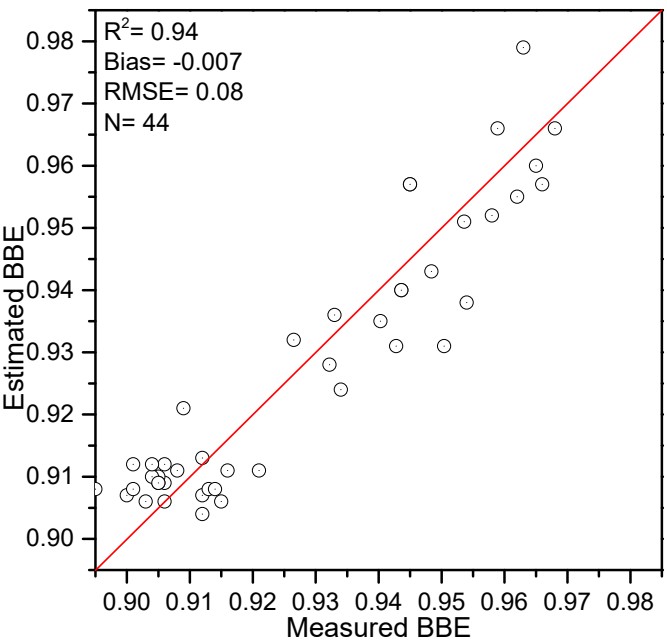

**Figure 3.** The scatterplot of BBE measured and BBE estimated by Equation (11) from the MODIS products.

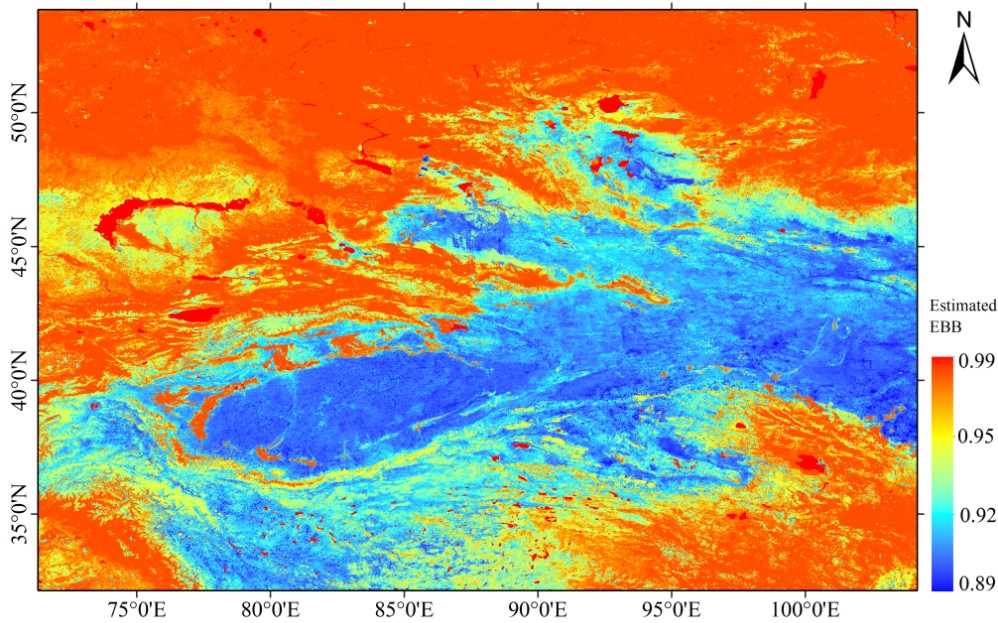

**Figure 4.** Estimated broadband emissivity map over Northwest of China.

### 4.3. Comparison with BBE of GLASS and Model Used Default Value

To further analyze the difference between this data and other products, we compared the estimated BBE in June 2018 with GLASS and land models (Noah) that used emissivity values by a look-up table. From Figure 5a, its clear to see that the GLASS BBE value (about 0.94) is higher than that estimated our estimated MODIS (0.90) over deserts when referring to Figure 4. Figure 6 shows histograms and scatterplot of the BBE of GLASS and our method generated over the Taklimakan Desert, the GLASS BBE is concentrated between 0.92 to 0.94, whereas our estimated range is 0.90 to 0.915, two peak BBE value at approximately 0.905 and 0.93, respectively. The GLASS BBE illustrates an inhomogeneous distribution in the eastern part of the Taklimakan Desert. The GLASS BBE of vegetated regions (oasis cropland and prairies) range from 0.97 to 0.98 and the BBE in sparse vegetation areas was about 0.96. The BBE estimated from MODIS over the above-mentioned land surface was higher than GLASS. Notably, the GLASS BBE of water bodies is the default (No-data) correspond to white, whereas the estimated BBE is 0.99. The land model used emissivity maps from Figure 5b, where the value was generated by using a look-up table and land-use type matching. Obviously, to see that each type of land cover has a unique value without variation and there is no gradient variation of vegetation and desert in spatial distribution. The BBE of Noah are 0.90 and 0.985 for barren or sparsely vegetated land and oasis cropland, respectively, which approximates our estimated BBE from MODIS. The land surfaces in arid regions are complex, varied and non-homogeneous. Thus, our estimated BBE is closer to reality.

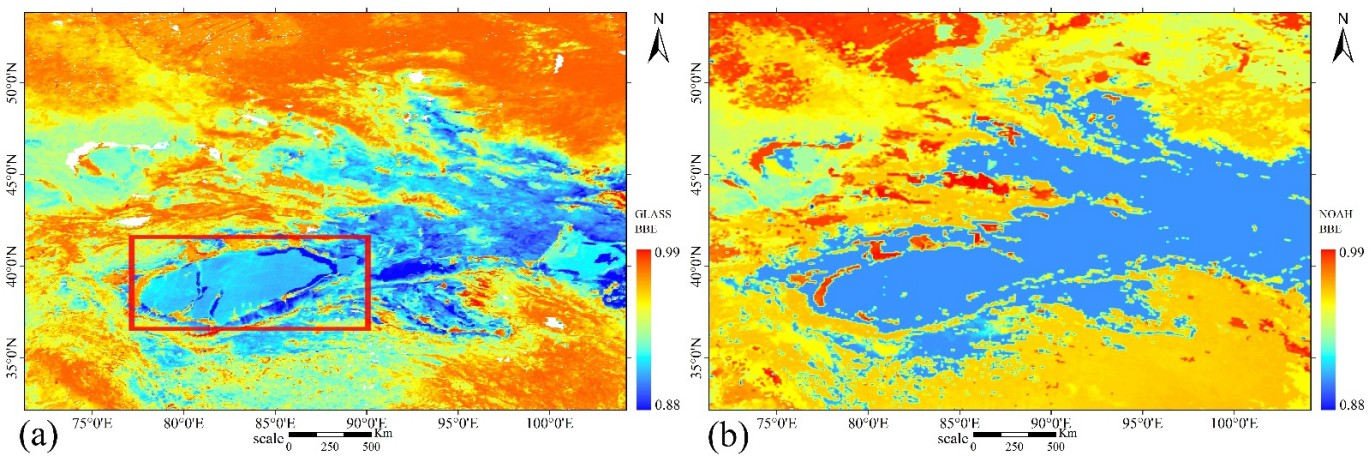

**Figure 5.** GLASS BBE data (**a**) and the emissivity derived from look-up table of Noah (**b**) over Northwest of China.

### 4.4. Analysis of the Relationship with Land Cover and Soil Moisture

Emissivity has a close relationship with land-use types and soil moisture [59,60]. To further investigate the spatial distribution of the correlation of BBE with land-use and soil moisture, we conducted a contrast analysis of the BBE distribution to the land cover map and soil moisture. Figure 7a displays each type of land-use over arid region, where dominant land-use types are desert and sparse vegetation. There exists a high correlation between the BBE and land-use types distribution characteristics refer to Figure 4. The barren soil (label: 200, 201, 202, according to the land cover code introduced in Appendix (Table A1) have a lower emissivity, with a value of approximately 0.90. Furthermore, the cropland (label: 20) and water body areas (label: 210, denoted in blue) are consistent with the BBE (denote in red). The BBE characterizes various features over vegetated areas and vegetation–desert transition zones, primarily due to emissivity from the most variable part of the TIR spectrum, especially vegetation areas. Further to say, the patterns of the BBE and land-use are very similar.

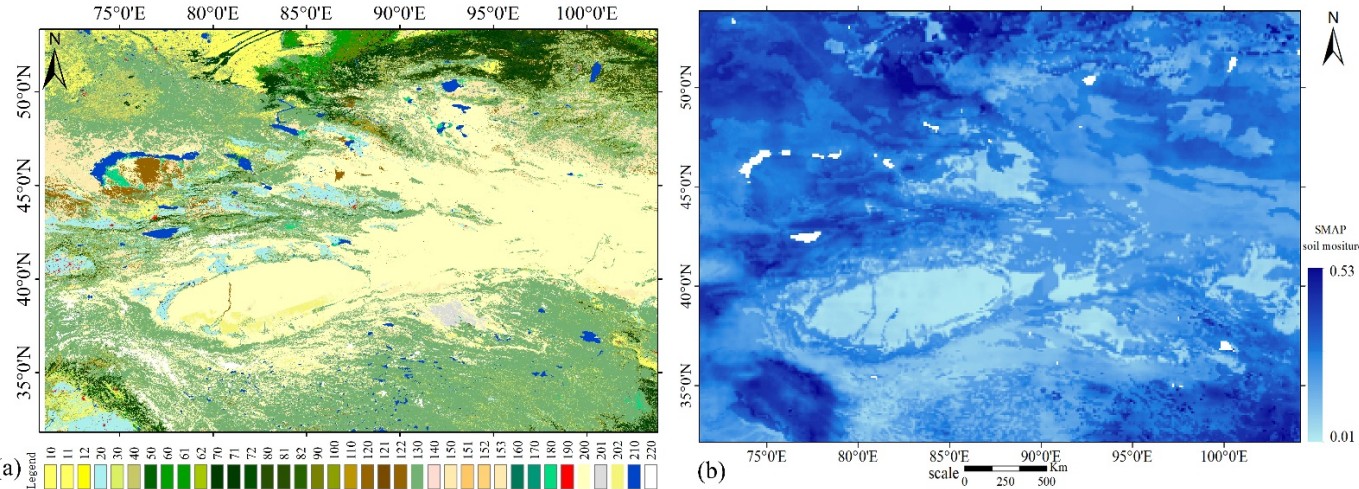

**Figure 6.** The scatterplot BBE of GLASS and estimated and histograms of the BBE of GLASS (blue line) and our method generated (red line) over TD (top left corner). The statistical area is the rectangular range in Figure 5**a**.

**Figure 7.** Land cover map (**a**) and volumetric moisture content of soil (**b**) over Northwest of China.

The influence of soil moisture in thermal infrared emissivity is a known fact [61]. The relationship BBE and soil moisture was investigated using the soil SMAP soil moisture products, as shown in Figure 7b, it is clear to see that barren soil has the lowest soil moisture (less than 0.05 m$^3$/m$^3$) such as TD and GD and the corresponding emissivity is the lowest, according to Figure 4, the pattern of which is consistent with soil moisture. The soil moisture of the Tianshan Mountains and in dense vegetation areas is a higher value, which corresponds to the lowest values of BBE. Figure 8 shows a scatterplot of BBE and soil moisture over the arid region, it illustrates that the BBE increases with increasing soil water content. According to the variation trend of BBE and soil moisture in spatial over TD, we conducted a natural logarithm function between BBE and soil moisture, while the $R^2$ is very low. BBE still changed to some extent in the region with very low soil moisture, this dependence is negligible, indicating that soil color and soil texture influenced the contrast emissivity. Nevertheless, oasis cropland has medium soil moisture, whereas the corresponding area has a higher BBE (0.985). The results demonstrate that the BBE was determined by both the surface wetness and vegetation conditions. Besides, due to lack of snow mountains, so it's hard to deduct a linear relationship over ice/snow land cover (Figure 7a denoted in white).

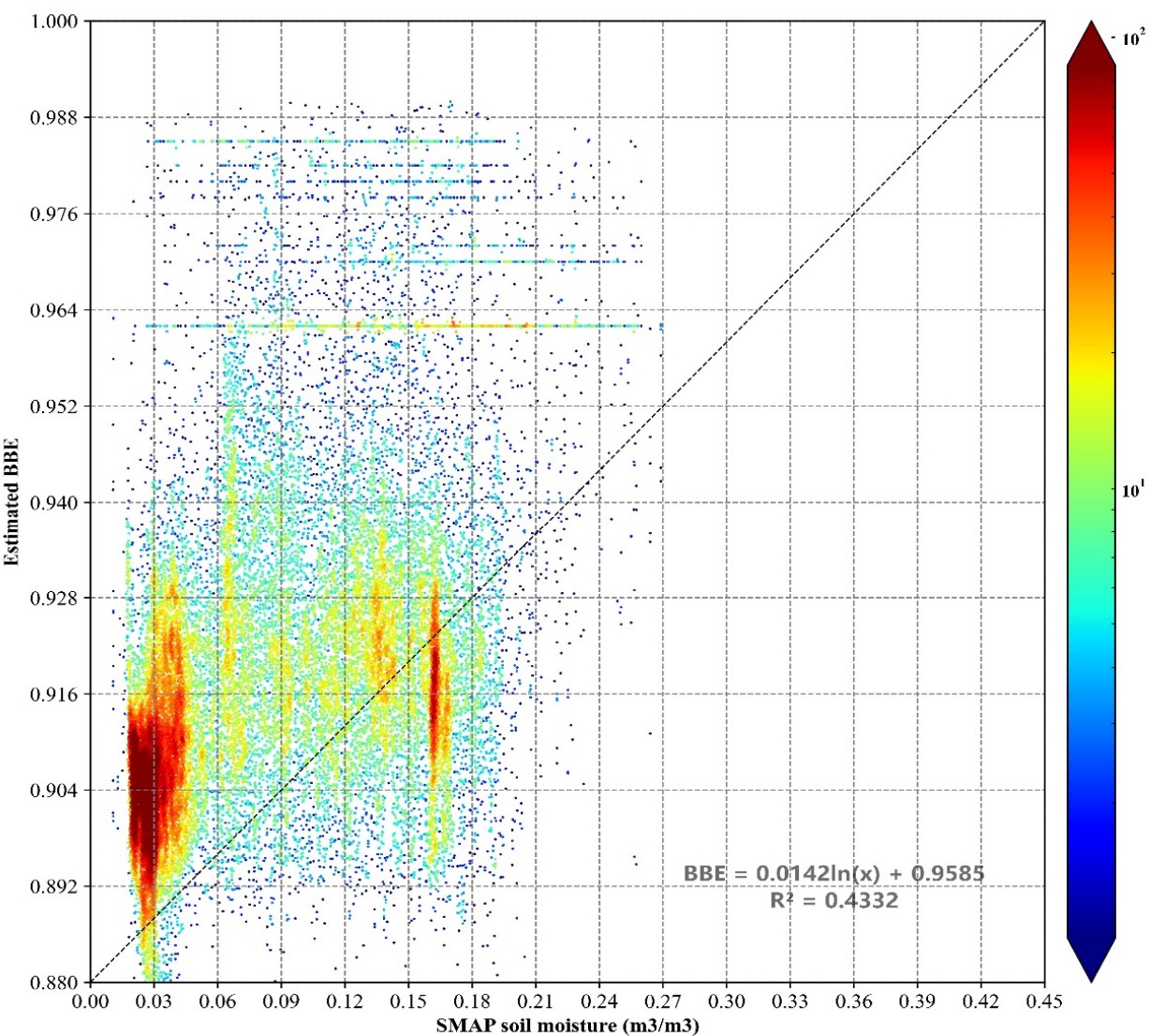

**Figure 8.** The scatterplot of estimated BBE and volumetric moisture content of soil over TD.

## 5. Discussion

Our proposed method's primary purpose is to compensate for the often-poor quality of MODIS emissivity products and improve the accuracy of BBE estimation over arid regions. Previous studies proposed methods that only use remote sensing data in different thermal infrared bands to estimate BBE and are not validated by spectral measurement. Considering the spatial differentiation of soil texture and FVC over arid regions, we propose to add LAI and reflectance into the linear fitting equation. Based on the field-measurement spectral and MODIS products to fit a linear equation to estimate BBE, the above results and analysis indicate that Equation (11) has the best performance; the primary reasons are that our equation provides more information on the land surface characterization. Our estimated BBE compared to GLASS and model used a default value, the BBE was lower than the GLASS over deserts and better agreed with ground measurements and the oasis-cropland BBE was higher than GLASS. The BBE described more detailed spatial variation than GLASS over vegetated area and vegetation–desert transition zones. In addition, the BBE used in the model depends entirely on the land-use types, it has static values on the same type without spatial differentiation, obviously, which is not in line with the actual situation. Moreover, our BBE value was close to 0.99 over water bodies but GLASS had no available data. Then, we analyzed the similar relationship of the BBE and land-use types. The BBE distributions of desert, water bodies and cropland were consistent with each type of land-use. The results indicated a high relationship between BBE and soil moisture. To sum up, our results show that BBE estimations with the proposed method have a high spatial dynamic range and facilitate accurate land surface BBE estimations in arid regions. Furthermore, this study proves that the emissivity variation with soil moisture, so we can try to add soil moisture as a variable into the method to further improve the accuracy of estimating BBE.

It is necessary to emphasize that this study is very local, and we did not measure the spectral over forest and cropland areas. This method may have uncertainties in estimating BBE. On one hand, we will take more measurements on forest, crop and urban areas, to supplement more spectral data over different types of land-use and try to extrapolate this method to other regions. On the other hand, we look forward to performing extensive tests of our retrieved BBE estimations in a land model in future studies.

## 6. Conclusions

The objective of this study was to explore a new method that uses different MODIS products with situ field-measured surface spectral data to fit linear regression equations for estimating the broadband emissivity in arid regions. We interpreted the physical mechanism of the multiple narrowband emissivity linear combination and we proposed three linear equations that include different MODIS products. First, we calculated field-measured spectral data and converted these data to BBE in the wavelength range of 8–14 μm, then, we extracted in situ values from MODIS products of the narrowband emissivity, reflectance and LAI, the coefficients of the established linear regression equations and were validated by field-measured emissivity data. The first equations had the lowest correlation; the $R^2$ was 0.83, and the RMSE and average bias were 0.17 and 0.14, respectively. The third equation involved reflectance and LAI products and its accuracy improved significantly. The reflectance of MODIS in channel 7 could be used to determine inhomogeneous soil texture, especially over barren soil land-use type. Thus, reflectance is necessary for estimating the BBE in arid regions. Equation (11) added LAI data based on Equation (11) and provided more vegetation information, which further improved the accuracy of BBE Even though Northwest of China is a region dominated by a vast desert, some oases exist, with agricultural areas, prairies and forests distributed in the northern and southern areas of the Tianshan Mountains, these vegetated areas have a higher BBE and are closely associated with the LAI. As a consequence, the BBE estimated with Equation (11) described detailed spatial variation over vegetated areas and vegetation–desert transition zones. Then, we compared the BBE estimated with Equation (11) to the GLASS BBE products, the GLASS BBE over the desert was higher than 0.94 and

was obviously overestimated compared to field measurements. The BBE estimated by our method ranged from 0.90 to 0.92 and thus agreed well with FTIR observations. A significant difference is that the real BBE for a body of water should be close to 1, where our estimated BBE was nearly 0.99, whereas the GLASS BBE had no available data. We demonstrated that the BBE was influenced by soil moisture, vegetation fraction and soil texture by analyzing the relationship between the BBE, land cover and soil moisture. Our estimation method comprehensively considers land surface conditions, and the method mitigates the variation and uncertainty in the range of land surface emissivity estimations in arid regions and showed a correlation with land-use type and soil moisture. This study provided a new perspective on estimating BBE from joint ground observation and satellite data, and the results of our comparative evaluation of the proposed method demonstrated its superior performance over previous methods. Furthermore, the framework developed in this study can easily be extended to other remote sensing data.

**Author Contributions:** Conceptualization, H.L. and Z.L.; methodology, H.L.; software, Z.G.; validation, H.L., Z.L. and H.Z.; formal analysis, C.J.; investigation, Y.L.; resources, A.M.; data curation, Z.G.; writing—original draft preparation, J.L.; writing—review and editing, Y.L.; visualization, Z.G.; supervision, A.M.; project administration, H.Z.; funding acquisition, J.L. All authors have read and agreed to the published version of the manuscript.

**Funding:** The research was supported by the National Natural Science Foundation of China (41801019, 41875023, 41675011, 41805075), the Research Foundation of Central Asia for Atmospheric Science (CAAS201811) and the National Key Research and Development Program of China (2018YFC1507105).

**Institutional Review Board Statement:** Not applicable.

**Informed Consent Statement:** Not applicable.

**Data Availability Statement:** The remote sensing data for this study is available on the website link in Section 2.2.

**Acknowledgments:** We thank Marko Mladenovic and anonymous reviewers for their insightful suggestions and thoughtful revision.

**Conflicts of Interest:** The authors declare no conflict of interest.

## Appendix A

**Table A1.** ESACCI GLC2015 land cover map classification index table.

| Label | land Cover Description |
|:---:|:---|
| 0 | No Data |
| 10 | Cropland, rainfed |
| 20 | Cropland, irrigated or post-flooding |
| 30 | Mosaic cropland (>50%)/natural vegetation (tree, shrub, herbaceous cover) (<50%) |
| 40 | Mosaic natural vegetation (tree, shrub, herbaceous cover) (>50%)/cropland (<50%) |
| 50 | Tree cover, broadleaved, evergreen, closed to open (>15%) |
| 60 | Tree cover, broadleaved, deciduous, closed to open (>15%) |
| 70 | Tree cover, needle-leaved, evergreen, closed to open (>15%) |
| 80 | Tree cover, needle-leaved, deciduous, closed to open (>15%) |
| 90 | Tree cover, mixed leaf type (broadleaved and needle-leaved) |
| 100 | Mosaic tree and shrub (>50%)/herbaceous cover (<50%) |
| 110 | Mosaic herbaceous cover (>50%)/tree and shrub (<50%) |
| 120 | Shrubland |
| 130 | Grassland |
| 140 | Lichens and mosses |
| 150 | Sparse vegetation (tree, shrub, herbaceous cover) (<15%) |
| 160 | Tree cover, flooded, fresh or brackish water |
| 170 | Tree cover, flooded, saline water |
| 180 | Shrub or herbaceous cover, flooded, fresh/saline/brackish water |
| 190 | Urban areas |
| 200 | Bare areas |
| 210 | Water bodies |
| 220 | Permanent snow and ice |

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
