# Peer review of "A New Linear Relation for Estimating Surface Broadband Emissivity in Arid Regions Based on FTIR and MODIS Products"

_remotesensing, doi:10.3390/rs13091686_

Round 1
Reviewer 1 Report
1.This is a very interesting article to research a new method for estimating surface broadband emissivity in Satellite image.
2.The proposed method needs narrowband emissivity, reflectance, and LAI products. The emissivity is derived from MODIS land surface temperature and emissivity products. However, the accuracy of MODIS image is only 1000m * 1000m, and the result of calculation is not accurate enough. If it could use Landsat to build the model, it will be better.
3.In order to make the methodology more universal, this paper only takes the arid area as an example to verify the model, and the subtropical area and the cold zone area should be added.
4.In addition, this paper only creates a model of land use in arid areas. From the perspective of land use types, this paper should increase the land use in urban built-up areas, agricultural land and forest areas, so as to make the methodology more universal.
Author Response
To Revierwer1
According to your suggestions, we have made some modifications to the article, including the introduction and the research area, and updated high-quality pictures. And professionally proofread by native English speakers. The quality of the manuscript has been greatly improved. Thank you very much for your advice!
Comments and Suggestions for Authors
1.This is a very interesting article to research a new method for estimating surface broadband emissivity in Satellite image.
- Response: Thank you for your encourage, we will improve this manuscript before publish.
2.The proposed method needs narrowband emissivity, reflectance, and LAI products. The emissivity is derived from MODIS land surface temperature and emissivity products. However, the accuracy of MODIS image is only 1000m * 1000m, and the result of calculation is not accurate enough. If it could use Landsat to build the model, it will be better.
- Response: Landsat has a higher resolution, actually, but which thermal infrared bandwidths (10.6-12.51 μm) can not cover the broadband emissivity (8-13.5 μm) we need. Cheng et al.(JGR,2017 ) developed a framework for estimating the 30 m thermal-infrared broadband emissivity from Landsat surface reflectance data, this method relies on ASTER and Landsat data by look-up table to retrieve the 30 m BBE over bare soils, this method not easily to be generalized to other vegetated areas. Our proposed method uses MODIS data to generate BBE that could reprocess the land surface model's static data in future work.
3.In order to make the methodology more universal, this paper only takes the arid area as an example to verify the model, and the subtropical area and the cold zone area should be added.
- Response: Your suggestion is insightful. This method based on FTIR field-measured surface spectral data and MODIS products, at present, we haven't observed it in subtropical and cold regions. But we can try to verify the method over subtropical and cold regions in subsequent work.
- In addition, this paper only creates a model of land use in arid areas. From the perspective of land use types, this paper should increase the land use in urban built-up areas, agricultural land and forest areas, so as to make the methodology more universal.
- Response: I agree with you. In the next work, we plan to take a measurement spectral on different land use types. The spectral signals obtained by remote sensing are averaged at pixel scale, we plan to use FTIR measure on grassland, cropland, urban built-up surface and water body. Thus we will get more accurate BBE over heterogeneous underlying surface.

Reviewer 2 Report
Manuscript ID: remotesensing-1154407
Title: A new method for estimating surface broadband emissivity in arid regions based on FTIR and MODIS products.
Comments and Suggestions for Authors: This study assessed a new method that uses different MODIS products with in situ field-measured surface spectral data to fit linear regression equations for estimating the broadband emissivity in arid regions. However, there are a few issues which the authors should first clarify before the paper can be considered publication.
- Section 2 the authors should add from where they download the data with the website link.
- Please add the importance of the study region; this part should be introduced in separated paragraph with more specific information, research background and previous references in the section of Introduction.
- Line 301-305 the authors can delete this paragraph. The content in the Discussion should be deployed based on the Section of the result or about the whole work from a higher perspective.
- In the discussion part the authors only made descriptions about the results. From my point, the authors should compare the results with previous work and add more recent references to add more robust proof for conclusions.
- The authors need to link their paper's analysis to papers that have previously appeared in Remote Sensing- MDPI. This is in line with a question of relevancy, i.e., whether Remote Sensing-MDPI readers will benefit maximally by a paper that may not strongly possess evidence of an impact and above all relevancy to the journal's contents.
- Please check out the grammar.
Author Response
To Revierwer2
According to your suggestions, we have made some modifications to the article, including the introduction and the research area, and updated high-quality pictures. And professionally proofread by native English speakers. The quality of the manuscript has been greatly improved. Thank you very much for your advice!
Comments and Suggestions for Authors: This study assessed a new method that uses different MODIS products with in situ field-measured surface spectral data to fit linear regression equations for estimating the broadband emissivity in arid regions. However, there are a few issues which the authors should first clarify before the paper can be considered publication.
Section 2 the authors should add from where they download the data with the website link.
Please add the importance of the study region; this part should be introduced in separated paragraph with more specific information, research background and previous references in the section of Introduction.
Response: We have added the website link to download the data used in this study. And then, we added a section to introduce the study region and more specific information, research background and previous references.
Line 301-305 the authors can delete this paragraph. The content in the Discussion should be deployed based on the Section of the result or about the whole work from a higher perspective.
In the discussion part the authors only made descriptions about the results. From my point, the authors should compare the results with previous work and add more recent references to add more robust proof for conclusions.
Response: We have deleted words in lines 301 to 305, this section has been reorganized, then compare to the previous study, also discuss the difference between our method and previous, and added some references.
The authors need to link their paper's analysis to papers that have previously appeared in Remote Sensing- MDPI. This is in line with a question of relevancy, i.e., whether Remote Sensing-MDPI readers will benefit maximally by a paper that may not strongly possess evidence of an impact and above all relevancy to the journal's contents.
Response: According your suggestion, we added 4 references published on Remote sensing- MDPI related to this study.
Please check out the grammar.
Response: We checked the grammar full text, and the manuscripts has been edited by native English speakers.

Reviewer 3 Report
The present paper proposes a relation to estimate the surface broadband emissivity (BBE) in semi-arid context from MODIS products such emissivity of band 29, 31 and 32 (MOD11B1), reflectance of band 7 (MOD09A1) and LAI (MOD15A2H) products. The relation is established based on 44 in situ measured values of broadband emissivity acquired with FTIR spectrometer on the Taklimakan desert hinterland (China).
The authors then generalize this relationship to a larger area encompassing the Taklimakan Desert by generating a BBE emissivity map. This map is then compared to the GLASS BBE product over the desert area. A short discussion is also conducted by a qualitative comparison with the ESA CCI GLC2015 land cover map classification and a sub-surface soil moisture map derived from SMAP products.
General comments:
The paper is relatively well written and clear enough. However, my opinion is that the authors did a probably interesting work, but in a quick and not very thorough way. Moreover, the study is very local and the authors should be more modest about its quality to be extrapolated to other regions, even arid ones.
The title indicates "a new method", in fact the authors propose a linear relationship calibrated and adapted to the desert region of Taklimakan in West China according to a set of surface measurements. Any application elsewhere would be extrapolation. Such relation already exist: as an example, Wang et al. 2005 propose different linear combination of same MODIS emissivity bands (29, 31 and 32) to estimate broadband emissivity from in situ measurements. A bibliographic work around the already existing relations using narrow band MODIS emissivity, in order to justify the establishment of a new relation, would be a plus.
A more adequate title could be: “A new linear relation for estimating surface broadband emissivity in the arid region of Taklimakan desert context (China) based on FTIR and MODIS products”
There is no reference to the date of the field acquisitions, nor to the date of the MODIS products used to map the BBE at the region scale. This is a problem when comparing a BBE value to soil moisture, which is highly variable over time. The discussion and conclusions should take this temporality into account.
I found equations 3 to 6 confused, in particular the index rating. Equations 4 and 5 are same.
The reader lacks information on the range of LAI associated with field measurements (FTIR) and MODIS data from the area. This range could give an idea of the range of validity of the relationship under vegetated surface conditions. Is there any vegetation among the 44 measurement points? In the same sense, I wonder why the authors did not choose the NDVI (less indirectly estimated index) instead of the LAI.
Section 4.3 comparison with other emissivity estimated maps: this section is not clear and does not convince me at all. The Landsat model is not model compared is not detailed? The legend Fig. 6b indicates Noah? The maps do not have the same footprints, nor the same colorbar which does not help the comparison. No indication of the area used to generate the histogram in Figure 7 comparing BBE values between GLASS and the present relationship. The area should be identified on one of the maps Fig5 or 6.
The analysis between the generated BBE map and the land use and surface moisture maps is too superficial. It is obvious that emissivity depends on the type of surface. A scatter plot by surface class would however show some diversity.
The relationship with soil moisture is also known, soil moisture content modify emissivity value but this relation is complex (Lesaignoux et al. 2013) and depend on soil type. However, the analysis proposed is very quick, and does not bring anything.
My conclusion, why not a new relationship adapted to this area that would offer a more precise product. But I would remain more modest when it comes to its field of validity. I would compare it with already established relationships. The compraison with the GLASS product would be interesting if there was a better understanding of what is being compared and what is at stake. The discussion dealing with the land cover map is only interesting if you use it to filter out the areas of invalidity of the relationship (water, dense vegetation, snow...). And finally, the link with surface moisture should only be made in a multi-temporal analysis way, which is not the case here.
References:
Wang, K.; Wan, Z.; Wang, P.; Sparrow, M.; Liu, J.; Zhou, X.; Haginoya, S. Estimation of Surface Long Wave Radiation and Broadband Emissivity Using Moderate Resolution Imaging Spectroradiometer (MODIS) Land Surface Temperature/Emissivity Products. Journal of Geophysical Research: Atmospheres 2005, 110, doi:https://doi.org/10.1029/2004JD005566.
Lesaignoux, A.; Fabre, S.; Briottet, X. Influence of Soil Moisture Content on Spectral Reflectance of Bare Soils in the 0.4–14 Μm Domain. International Journal of Remote Sensing 2013, 34, 2268–2285, doi:10.1080/01431161.2012.743693.
Author Response
To Revierwer3
According to your suggestions, we have made some modifications to the article, including the introduction and the research area, and updated high-quality pictures. And professionally proofread by native English speakers. The quality of the manuscript has been greatly improved. Thank you very much for your advice, you are academically rigorous, a thumbs-up for you!
Comments and Suggestions for Authors
The present paper proposes a relation to estimate the surface broadband emissivity (BBE) in semi-arid context from MODIS products such emissivity of band 29, 31 and 32 (MOD11B1), reflectance of band 7 (MOD09A1) and LAI (MOD15A2H) products. The relation is established based on 44 in situ measured values of broadband emissivity acquired with FTIR spectrometer on the Taklimakan desert hinterland (China).
The authors then generalize this relationship to a larger area encompassing the Taklimakan Desert by generating a BBE emissivity map. This map is then compared to the GLASS BBE product over the desert area. A short discussion is also conducted by a qualitative comparison with the ESA CCI GLC2015 land cover map classification and a sub-surface soil moisture map derived from SMAP products.
General comments:
- The paper is relatively well written and clear enough. However, my opinion is that the authors did a probably interesting work, but in a quick and not very thorough way. Moreover, the study is very local and the authors should be more modest about its quality to be extrapolated to other regions, even arid ones.
Response: Thank you for your suggestion. This method based on FTIR field-measured surface spectral data and MODIS products. At present, we haven't observed it on subtropical and cold regions. But we can try to verify the method over subtropical and cold region in next work.
- The title indicates "a new method", in fact the authors propose a linear relationship calibrated and adapted to the desert region of Taklimakan in West China according to a set of surface measurements. Any application elsewhere would be extrapolation. Such relation already exist: as an example, Wang et al. 2005 propose different linear combination of same MODIS emissivity bands (29, 31 and 32) to estimate broadband emissivity from in situ measurements. A bibliographic work around the already existing relations using narrow band MODIS emissivity, in order to justify the establishment of a new relation, would be a plus.
Response: I agree with your viewpoints, previous studies has proposed different linear combination of same MODIS emissivity bands to estimate BBE, but have less observation to validate the models, and MODIS emissivity product (MOD11B1/MOD11A1) quality is poor in vegetated areas. We measured the spectral on different types of arid land surface, and added surface reflectance and LAI into liner regression. Reflectance represents the spatial differences of bare soil, LAI could reflect vegetation information, more surface information is helpful to improve the estimation accuracy of heterogeneous surface.
- A more adequate title could be: “A new linear relation for estimating surface broadband emissivity in the arid region of Taklimakan desert based on FTIR and MODIS products”
Response: Thank your suggestion, we have modified the tile as “A new linear relation for estimating surface broadband emissivity in the arid region of Taklimakan desert based on FTIR and MODIS products”. The new tile is more aligned with the content of the article.
- There is no reference to the date of the field acquisitions, nor to the date of the MODIS products used to map the BBE at the region scale. This is a problem when comparing a BBE value to soil moisture, which is highly variable over time. The discussion and conclusions should take this temporality into account.
Response: We add the date of MODIS and field-measurement in the context. Soil moisture does have a significant effect on emissivity indeed, in arid areas, the frequency of rainfall is low and evaporation is rapid, so the soil in arid areas stays dry for a long time, soil moisture variation very slowly at at all seasons.considering the climatic characteristics of the arid area, we chose this method to test in the arid area. In the Noah model, focring data provides meteorological condition (precipitation, snow cover fraction et al.), in case of precipitation, the model will recalculates the surface emissivity based on static surface data (include BBE). We should add more discussion about the influence of soil moisture on BBE.
- I found equations 3 to 6 confused, in particular the index rating. Equations 4 and 5 are same.
Response: We checked the Equations 3 to 6, and deleted Equation 5, and carefully introduce the Equations 3 to 6 in the context.
- The reader lacks information on the range of LAI associated with field measurements (FTIR) and MODIS data from the area. This range could give an idea of the range of validity of the relationship under vegetated surface conditions. Is there any vegetation among the 44 measurement points? In the same sense, I wonder why the authors did not choose the NDVI (less indirectly estimated index) instead of the LAI.
Response: We added some information of MODIS data and FTIR field-measured land surface in the context. Some of the sites are covered with sparse vegetation, we not measured the vegetation leaf directly but measured the surface. If we observe the vegetation directly, the uncertainty of the measurement results is so great that the data is not reliable, the nearby vegetation still has an influence on the observation, which is higher emissivity than that of the desert. From Figure1 we can see that observation sites around less vegetation. LAI can can reflect the growth of vegetation, water content, we make a normalization of LAI before used. We have read some research papers that NDVI can be used to estimate emissivity indeed, MODIS NDVI (MOD13Q1) products is an 16-day composite datasets, but we used other MODIS products (MOD15A2H, MOD11B1, MOD09A1) are 8-day temporal resolution, different temporal resolution products would impact the accuracy of estimating BBE.
- Section 4.3 comparison with other emissivity estimated maps: this section is not clear and does not convince me at all. The Landsat model is not model compared is not detailed? The legend Fig. 6b indicates Noah? The maps do not have the same footprints, nor the same colorbar which does not help the comparison. No indication of the area used to generate the histogram in Figure 7 comparing BBE values between GLASS and the present relationship. The area should be identified on one of the maps Fig5 or 6.
Response: Landsat data has higher resolution, but which thermal infrared bandwidths (10.6-12.51 μm) can not cover the broadband of BBE (8-14 μm), Landsat requires complex preprocessing to obtain infrared bands. Landsat data may not easy to extend to larger areas. We redraw the Figure6 use same colorbar for easier to tell the difference. And then we marked the three types land surface observation sites of Figure7 on Figure6.
- The analysis between the generated BBE map and the land use and surface moisture maps is too superficial. It is obvious that emissivity depends on the type of surface. A scatter plot by surface class would however show some diversity.
Response: We added a deeper analysis of the relationship between soil moisture and emissivity, and gave the scatter diagram and correlation coefficient
- The relationship with soil moisture is also known, soil moisture content modify emissivity value but this relation is complex (Lesaignoux et al. 2013) and depend on soil type. However, the analysis proposed is very quick, and does not bring anything.
Response: we added the analysis of the relation between BBE and soil moisture and added this paper in references.
- My conclusion, why not a new relationship adapted to this area that would offer a more precise product. But I would remain more modest when it comes to its field of validity. I would compare it with already established relationships. The compraison with the GLASS product would be interesting if there was a better understanding of what is being compared and what is at stake. The discussion dealing with the land cover map is only interesting if you use it to filter out the areas of invalidity of the relationship (water, dense vegetation, snow...). And finally, the link with surface moisture should only be made in a multi-temporal analysis way, which is not the case here.
Response: Your suggestion is insightful. We think this relationship needs to be based on observation, we should take more field-measurement over different land use, to build relationships estimate global BBE and compare with GLASS BBE. Our plan not only to estimate BBE, the main purpose is to develop high-resolution land surface parameters for use in land surface models. As mentioned in the paper, the accuracy of the land surface parameters used in the current land surface model is very low and it is difficult to match with the actual surface status, the BBE generated by our method has been re-process into static data of Land model, and test it in future. Therefore, according to your suggestion we will also continue to refine the BBE in more tended estimation and validation.
References:
Wang, K.; Wan, Z.; Wang, P.; Sparrow, M.; Liu, J.; Zhou, X.; Haginoya, S. Estimation of Surface Long Wave Radiation and Broadband Emissivity Using Moderate Resolution Imaging Spectroradiometer (MODIS) Land Surface Temperature/Emissivity Products. Journal of Geophysical Research: Atmospheres 2005, 110, doi:https://doi.org/10.1029/2004JD005566.
Lesaignoux, A.; Fabre, S.; Briottet, X. Influence of Soil Moisture Content on Spectral Reflectance of Bare Soils in the 0.4–14 Μm Domain. International Journal of Remote Sensing 2013, 34, 2268–2285, doi:10.1080/01431161.2012.743693.
We added these references to manuscripts.
